# Natural Variation in Adventitious Rooting in the Alpine Perennial *Arabis alpina*

**DOI:** 10.3390/plants9020184

**Published:** 2020-02-03

**Authors:** Priyanka Mishra, Adrian Roggen, Karin Ljung, Maria C. Albani

**Affiliations:** 1Institute for Plant Sciences, University of Cologne, Zülpicher Str. 47B, 50674 Cologne, Germanyroggen@mpipz.mpg.de (A.R.); 2Max Planck Institute for Plant Breeding Research, Carl-von-Linné-Weg 10, 50829 Cologne, Germany; 3Cluster of Excellence on Plant Sciences “From Complex Traits towards Synthetic Modules”, 40225 Düsseldorf, Germany; 4Umeå Plant Science Centre, Department of Forest Genetics and Plant Physiology, Swedish University of Agricultural Sciences, 90736 Umeå, Sweden; Karin.Ljung@slu.se

**Keywords:** *Arabis alpina*, alpine, perennial, adventitious rooting, clonal growth, auxin

## Abstract

Arctic alpine species follow a mixed clonal-sexual reproductive strategy based on the environmental conditions at flowering. Here, we explored the natural variation for adventitious root formation among genotypes of the alpine perennial *Arabis alpina* that show differences in flowering habit. We scored the presence of adventitious roots on the hypocotyl, main stem and axillary branches on plants growing in a long-day greenhouse. We also assessed natural variation for adventitious rooting in response to foliar auxin spray. In both experimental approaches, we did not detect a correlation between adventitious rooting and flowering habit. In the greenhouse, and without the application of synthetic auxin, the accession Wca showed higher propensity to produce adventitious roots on the main stem compared to the other accessions. The transcript accumulation of the *A. alpina* homologue of the auxin inducible *GH3.3* gene (*AaGH3.3*) on stems correlated with the adventitious rooting phenotype of Wca. Synthetic auxin, 1-Naphthaleneacetic acid (1-NAA), enhanced the number of plants with adventitious roots on the main stem and axillary branches. *A. alpina* plants showed an age-, dosage- and genotype-dependent response to 1-NAA. Among the genotypes tested, the accession Dor was insensitive to auxin and Wca responded to auxin on axillary branches.

## 1. Introduction

Arctic and alpine environments are characterized by low temperatures, strong desiccating winds, rocky soils with unstable substrates and short and unpredictable growing seasons that limit plant growth. These conditions can make a great difference to plant fitness by reducing the survival of young seedlings or by hampering successful flowering, fruiting and germination [1,2,3]. To distribute risk and survive in severe habitats at high altitudes, many alpine species reproduce not only through seeds, but also clonally [1,3]. In this respect, the vast majority of the alpine species are perennials and can bypass sexual reproduction in years when seed production is unreliable [4,5]. In the Brassicaceae family, *Arabidopsis thaliana*, as an annual species, has not been reported to clonally propagate in nature [6]. Many perennial relatives of *A. thaliana,* however, propagate clonally through stolons (e.g., in *Arabidopsis halleri*) and rhizomes (e.g., in *Arabidopsis lyrata*) in parallel to their self-incompatible mating system [7]. The alpine perennial *Arabis alpina* does not produce special clonal structures, and most *A. alpina* populations in North Europe are inbreeders [8,9,10,11]. However, population landscape studies have indicated the presence of clonal individuals, suggesting that, in some habitats, *A. alpina* follows a mixed clonal-sexual reproductive strategy similar to other alpine perennials [12].

Alpine species often do not produce special clonal organs but can propagate using adventitious roots produced on stems creeping on the ground (probably a consequence of the winter snow) [3]. The ability of plant parts to develop adventitious roots has been exploited in horticulture such that several techniques promoting clonal propagation by cuttings, layering or tissue culture rely on a successful adventitious root formation. The phytohormone auxin has also been used as a rooting agent, since it induces development of adventitious roots in stem cuttings [13,14]. The molecular mechanisms regulating adventitious rooting have been explored in *A. thaliana*, as adventitious roots can develop on the hypocotyls of seedlings grown on media after removal of roots or dark to light shifts in vertically growing seedlings [15,16]. Mutants in *A. thaliana*, such as *superroot1* (*sur1*), have demonstrated a correlation between the excessive production of adventitious roots and enhanced free and conjugated auxin levels [17]. 

Here, we explored *A. alpina* as a model to study the process of adventitious rooting. We characterized the ability of *A. alpina* accessions to produce adventitious roots in greenhouse conditions and developed a foliar auxin spray approach to induce adventitious roots on intact plants. Similar to other studies, 1-NAA enhanced the number of plants with adventitious roots in a dose-dependent manner. Both approaches, resulted in the identification of natural variation for adventitious rooting among *A. alpina* accessions.

## 2. Results

### 2.1. The Wca Accession Has Higher Propensity to Produce Adventitious Roots on the Main Stem

To test for natural variation in adventitious rooting among *A. alpina* accessions, we grew Pajares (Paj), Dorfertal (Dor), Totes Gebirge (Tot) and West Carpathians (Wca) plants in a long-day greenhouse (n = 45). The accessions Dor, Tot and Wca can flower in greenhouse conditions, whereas the accession Paj does not because it requires cold treatment to flower (Appendix A; [18]). To ensure that flowering does not affect the ability of *A. alpina* accessions to produce adventitious roots, we measured flowering time and included the flowering time mutant *perpetual flowering 1* (*pep1-1*) in our analysis. The *pep1-1* mutant, which was previously derived after the mutagenesis of the Paj accession, flowers without cold treatment and therefore allowed us to assess adventitious rooting between a flowering (*pep1-1*) and a non-flowering genotype (Paj) in a similar genetic background ([18]; Appendix A). The number of plants with adventitious roots on the hypocotyl, the main stem and the axillary branches, were scored six weeks after sowing. At this developmental stage, the accessions did not produce adventitious roots on the axillary branches. However, we observed differences in the frequency of plants that developed adventitious roots on the hypocotyl or on the main stem (Figure 1, Appendix A). 

On the main stem, all Wca plants produced adventitious roots in at least one internode, whereas none of Tot and only a few of Dor plants did (Figure 1a,b, Appendix A). On the hypocotyl, *pep1-1* showed increased potential to develop adventitious roots compared to Paj, as well as compared to Dor, Tot and Wca (Figure 1a,c). Since we observed differences in adventitious rooting among Dor, Tot, Wca and *pep1-1* plants, although all these genotypes flower in greenhouse conditions, we conclude that there is no link between flowering behaviour and adventitious rooting. 

We then compared the transcript levels of the auxin conjugating *Gretchen Hagen 3* (*GH3*) genes, which were previously shown to regulate adventitious rooting on *A. thaliana* hypocotyls [19]. In stems of six-week-old *A. alpina* plants, *AaGH3.3* transcript levels were significantly higher in Wca compared to the other genotypes (Figure 2). GH3 enzymes in *A. thaliana* have been proposed to regulate endogenous auxin content by conjugating amino acids to IAA [20]. In our study, we detected no differences in the endogenous free IAA levels in stems between genotypes (Figure 2c). This result is in agreement with previous findings showing that GH3 enzymes do not control adventitious rooting by modulating free IAA content [19]. Overall, these results suggest that *A. alpina* accessions show natural variation in adventitious rooting and that *AaGH3.3* expression correlates with higher competence in adventitious rooting in the Wca accession.

### 2.2. Auxin Spray Induces Adventitious Roots in A Dosage-, Age- and Genotype-Dependent Manner

To induce adventitious roots in a more robust way, we sprayed intact plants with the synthetic auxin 1-NAA. We first optimized the protocol using only two genotypes, Paj and the flowering time mutant *pep1-1*. As both genotypes developed adventitious roots on the hypocotyl without the application of auxin, the presence of adventitious roots after auxin spray was scored only in the axillary branches and internodes on the main stem. At concentrations of 0 µM, 10 µM, 20 µM, 50 µM and 100 µM, 1-NAA was applied once using spray on sets of 10 plants. A clear leaf-curling phenotype was observed in a dosage-dependent manner (Appendix A). To follow the adventitious rooting phenotype, we scored the presence of adventitious roots on the main stem and axillary branches at one, two, three and five weeks after the application of 1-NAA. To specifically record the effect of auxin, the frequency of internodes/axillary branches with an adventitious root was followed only in the metameres present before spraying. Young seedlings grown only for three weeks in a long-day greenhouse did not develop adventitious roots in response to 1-NAA due to the absence of a definitive main stem and axillary branches (data not shown). However, six- and eight-week-old plants developed adventitious roots in a dosage-dependent manner following auxin spray (Figure 3; Appendix A). Adventitious roots were visible on the plants two weeks after auxin spray. Higher concentrations of auxin resulted in a higher portion of axillary branches or internodes on the main stem occupied with adventitious roots (Figure 3). 

However, the frequency of axillary branches with adventitious roots was higher compared to adventitious rooting internodes on the main stem (Figure 3f–j). Plant age clearly influenced the ability of plants to respond to auxin and develop adventitious roots (Appendix A). Eight-week-old sprayed plants responded more to auxin spray and developed adventitious roots even after the application of lower concentrations of auxin (Figure 3b,g). This was evident for both Paj and *pep1-1* plants. These results suggest that *A. alpina* shows a dosage-dependent response to adventitious rooting in an age-dependent manner.

We then tested the effect of auxin spray on the other accessions. Since some of these accessions flowered earlier than eight weeks in long days, we sprayed plants at the age of six weeks (Appendix A) [21]. Interestingly, Wca plants developed adventitious roots on the main stem at a higher frequency than the other accessions irrespective of whether they are sprayed with auxin or mock (Figure 4a–e, Appendix A). The accession Wca, however, responded to 1-NAA on the axillary branches (Figure 4f–j, Appendix A). The application of auxin to Tot plants promoted the formation of adventitious roots only in the axillary branches (Figure 4a–j, Appendix A). The accession Dor was insensitive to 1-NAA and did not produce adventitious roots on the main stem and axillary branches after the application of auxin (Figure 4a–j, Appendix A). From these results, we conclude that auxin spray induces adventitious roots in a genotype-dependent manner which differs between the main stem and axillary branches (Appendix A).

## 3. Discussion

Clonal growth is favoured in stressful habitats [22,23]. Many alpine species are able to clonally propagate when environmental conditions during flowering are unfavourable [24]. *A. alpina* plants develop adventitious roots on axillary branches, which can sustain the clonal growth of the plant through the formation of daughter ramets in natural populations [12,25]. Environmental conditions and the mechanisms that might induce adventitious rooting in certain *A. alpina* populations are unknown. We showed that adventitious rooting in *A. alpina* is not influenced by the flowering habit. We, nevertheless, detected variation between accessions on their ability to produce adventitious roots on the hypocotyl, the main stem and axillary branches. The accession Wca showed the highest propensity to produce adventitious roots on the main stem, although it contained a similar level of free auxin compared to the other accessions. Wca might be highly responsive to the endogenous auxin due to the higher expression of the *A. alpina* homologue of *GH3.3*, which has been shown in *A. thaliana* to regulate adventitious rooting through the conjugation of jasmonic acid [19]. Measurements of conjugated and free jasmonic acid would further clarify the molecular mechanisms that regulate adventitious rooting in *A. alpina* accessions.

Foliar spray with auxin or other rooting agents is an established method to induce adventitious roots in agricultural industries, and it has been proven to be the most efficient way of hormone delivery in stem cuttings compared to methods such as shoot-tip drench or stem injection [26,27,28]. Due to industrial interest around perennial species, the molecular mechanism regulating adventitious root formation has been explored using stem cuttings [29,30]. These studies have focused on adventitious rooting on stem cuttings with and without the addition of exogenous factors including auxin [30,31]. However, for basic research purposes, the use of stem cuttings is not the best experimental system, as wounding itself induces adventitious root formation and does not provide a feasible control system [32,33,34]. Here, we explored the effect of foliar auxin spray on intact plants. Similar to studies using cuttings as an experimental system, we showed that the application of auxin on intact plants had a dosage-dependent effect on adventitious rooting [35,36]. This result suggests that auxin spray in intact plants may be a promising tool to study adventitious rooting. However, in our study, concentrations less than 50 µM were not sufficient to induce the emergence of adventitious roots, suggesting that the delivery of auxin to intact plants is not as efficient compared to cuttings [37]. We identified an easy-to-root (Wca) and a difficult-to-root (Dor) accession in *A. alpina.* Similar variations among genotypes in response to auxin were reported in crop species such as apple, eucalyptus and cashew [38,39,40,41,42]. Thus, we can conclude that foliar spray effectively supports previous findings on the age-dependent and genotype-specific adventitious root formation in response to auxin [43,44]. 

In summary, we used two methods to screen for natural variation for adventitious rooting in *A. alpina*. The easy-to-root accession (Wca) and the difficult-to-root accession (Dor) provide the starting tool to dissect the molecular mechanisms regulating clonal growth through adventitious rooting in *A. alpina*. The identification of additional accessions with the distinct ability to produce adventitious roots will pave the path for understanding the molecular mechanisms contributing to adaptation in alpine environments.

## 4. Materials and Methods 

### 4.1. Plant Material and Growth Conditions 

The accessions Pajares (Paj), Dorfertal (Dor), Totes Gebirge (Tot) and West Carpathians (Wca) have been previously characterized for their flowering behaviour and seed traits [21,25]. The *pep1-1* mutant was identified during a mutagenesis screen of the accession Paj and has also been described previously by the authors of [18]. The number of plants scored in each experiment is mentioned in the results and figure legends. Plants were grown in soil in a controlled environment greenhouse in long days (LD; 16-h light and 8-h dark; fluence: ~150 µmol m^−2^ s^−1^) and a temperature of about 22 °C. Scoring for the presence of adventitious roots was performed six or eight weeks after seeds were sown on soil. The presence of adventitious roots was scored in the hypocotyl, in the internodes on the main stem and in the axillary branches. To control for other developmental traits, the number of days to flower and number of leaves at flowering were also recorded.

### 4.2. Gene Expression Analysis

For gene expression analysis, plants were grown for six weeks in a long-day-controlled environment greenhouse and stems were harvested, removing leaves, petioles and axillary buds. For each sample, the whole stem of seven plants, excluding the axillary buds, were pooled. Total RNA was extracted using the RNeasy Plant Mini kit (Qiagen) and DNA was removed using the DNA*-free* kit (Invitrogen), according to the manufacturer´s protocol. DNase-treated RNA (2 μg) was used to synthesize cDNA through reverse transcription with SuperScript II and oligo dT primer (18b) according to the manufacturer’s instructions, and 3 μL of a cDNA solution (1:5.5 dilution) was used as a template for each quantitative PCR reaction (qPCR). The CFX Connect Real-Time PCR Detection System (Bio-Rad) and the iQ SYBR Green Supermix (Bio-Rad) were used to perform the real-time quantitative PCR analyses using the following primers: AaGH3.3F: 5′-ACAATTCCGCTCCACAGTTC-3′ and AaGH3.3R: 5′-CAAGTAACACCGATGCGTTC-3′; AaGH3.6F: 5′-CACCTTGTTCCGTTCGATG-3′ and AaGH3.6R: 5′-TACCATTCATGCAAAGCTCC-3′; AaPP2AF: 5′-AGTATCGCTTCTCGCTCCAG-3′ and AaPP2AR: 5′-AACCGGTTGGTCGACTATTG-3′. The PCR programme for all primers was 95 °C for 3 min, followed by 40 cycles of 95 °C for 15 s and 60 °C for 60 s, then a melting curve (55–95 °C with a heating rate of 0.5 °C/5 s). Quantification of *AaGH3.3* and *AaGH3.6* expression was normalized using *AaPP2A* mRNA levels as an internal control for each sample. Values represent the average of three biological replicates and error bars the standard deviation of the mean.

### 4.3. Free IAA Quantification

Plant material (around 15 mg fresh weight) was purified as previously described [45]. The same samples were used for the quantification of free IAA and gene expression analysis. Before homogenization and extraction, 500 pg ^13^C_6_-IAA (Indole-3-acetic acid) of internal standard was added to each sample. Free IAA was quantified in the purified samples using combined gas chromatography—tandem mass spectrometry. The mean and the standard deviation represent three biological replicates. 

### 4.4. Auxin Spray

To artificially induce adventitious roots, plants were sprayed with 0 µM, 10 µM, 20 µM, 50 µM and 100 µM of the auxin analogue 1-Naphthaleneacetic acid (1-NAA, Sigma Aldrich) dissolved in 0.1% (*v*/*v*) DMSO. Tween-20 was added to the solutions at 0.2% (*v*/*v*) as a surfactant, and 1-NAA solutions were always prepared fresh before auxin spray and a constant volume of 50 mL of solution was sprayed per 10 plants. Following auxin spray, plants were kept in the controlled greenhouse and scored one, two, three and five weeks after spray. Adventitious roots were scored on the internodes of the main stem and on the axillary branches. The percentage of internodes on the main stem and on the axillary branches with adventitious roots were calculated with respect to the internodes present on each plant before spraying.

### 4.5. Statistical Analyses

A one-way analysis of variance (ANOVA) followed by Tukey’s multiple comparison post hoc-test with Bonferroni correction was used to test for significant differences (*P* < 0.05) for the qPCR and endogenous IAA quantification (Figure 2). These tests were carried out in R (Version 3.4.3).

Multifactorial ANOVA combined with post-hoc Bonferroni corrections were carried out for data in Figure 3 and Figure 4 to determine whether adventitious rooting on the branches and main stem was significantly (corrected *p* < 0.05) different. For Figure 3, our statistical model included genotype, age, concentration of auxin and their interactions as fixed effects. The statistical model for Figure 4 consisted of the genotype, concentration of auxin and their interaction as fixed effects.

## Figures and Tables

**Figure 1 plants-09-00184-f001:**
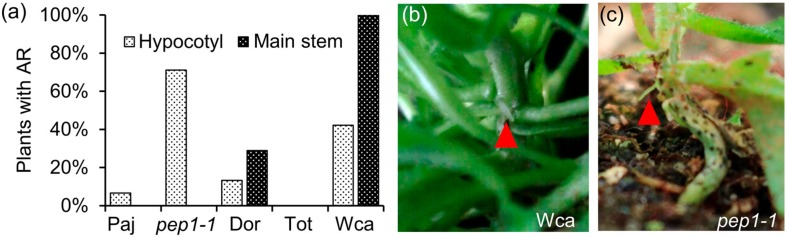
Natural variation for adventitious rooting in *A. alpina*. (**a**) Percentage of plants with adventitious roots on the hypocotyl and on the main stem in the *A. alpina* accessions Paj, Dor, Tot and Wca, and the *pep1-1* mutant. Plants did not have adventitious roots in the axillary branches. Forty-five plants were scored for each accession. In long days, adventitious roots (red arrowhead) are indicated on the (**b**) main stem of Wca and (**c**) hypocotyl of *pep1-1* plants.

**Figure 2 plants-09-00184-f002:**
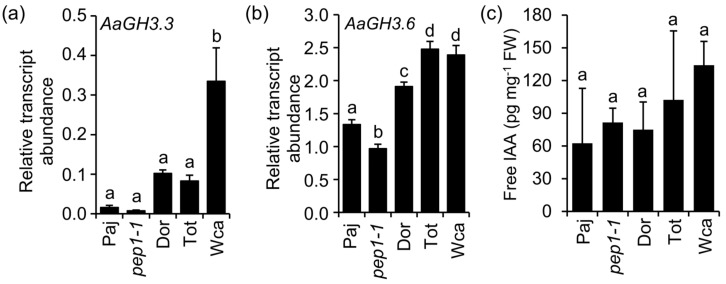
Transcript abundance of *A. alpina* homologues of *GH3.3* and *GH3.6* and free IAA content in stems. (**a**) *AaGH3.3*, (**b**) *AaGH3.6* and (**c**) endogenous free IAA levels. The main stem was harvested from six-week-old Paj, *pep1-1*, Dor, Tot and Wca plants. Stems from seven plants were pooled for each biological replicate. Results represent the average of three biological replicates. A one-way analysis of variance (ANOVA) followed by Tukey’s multiple comparison post *hoc*-test with Bonferroni correction was used to determine significantly different samples (*p* < 0.05) as presented in Appendix A. Error bars indicate SD of three biological replicates. FW denotes fresh weight.

**Figure 3 plants-09-00184-f003:**
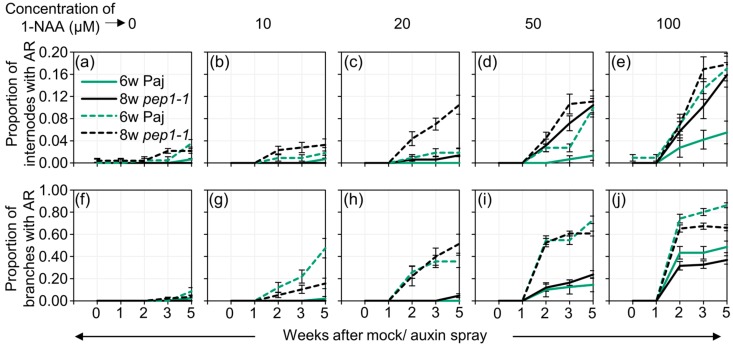
Auxin spray induces adventitious roots in a dosage- and age- dependent manner. Proportion of (**a**–**e**) internodes and (**f**–**j**) axillary branches with adventitious roots after the application of 0 µM, 10 µM, 20 µM, 50 µM and 100 µM 1-NAA relative to before spraying in six- (6w) and eight-week-old (8w) Paj and *pep1-1* plants. Ten plants were scored for each treatment. Plants were scored before spraying and one, two, three and five weeks after spraying. The data and statistical analyses are tabulated in Appendix A.

**Figure 4 plants-09-00184-f004:**
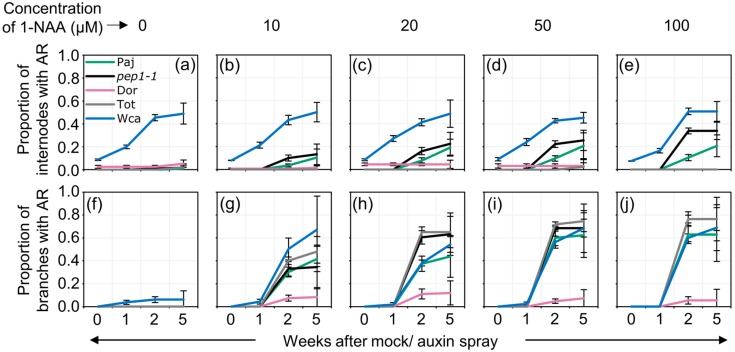
Auxin spray induces adventitious roots in a dosage- and genotype- dependent manner. Paj, *pep1-1*, Dor, Tot and Wca were sprayed after six weeks in LD with 0 µM, 10 µM, 20 µM, 50 µM and 100 µM 1-NAA. Nine plants were scored for each treatment. Plants were scored before spraying and one, two and five weeks after auxin spray. (**a**) Proportion of internodes on the main stem and (**b**) axillary branches occupied with adventitious roots after 1-NAA spray relative to before spray. The data and statistical analyses are tabulated in Appendix A. A second replicate is presented in Appendix A.

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
