# Peer review of "Natural Variation in Adventitious Rooting in the Alpine Perennial Arabis alpina"

_plants, 2020, doi:10.3390/plants9020184_

Round 1
Reviewer 1 Report
This paper presents a study of adventitous rooting in the perennial alpine Arabis alpina. The combination of observation of natural growth together gene transcription and effects of auxin application is commendable. However there are serious flaws in the presentation and interpretation of analyses.
The Abstract claims (line 25) that the Wca accession is auxin-resistant however the resuts show a response to auxin on auxillary branches. The lack of response on the main stem could be because it already expresses high level. Perhaps the comment on line 25 was meant for Dor accession which does appear resistant.
Sample sizes (number of plants grown, measured, sprayed etc) need to be made clear throughout the methods e.g. line 64-65, fig 1 caption, lines 104, 185, 213
Fig 2 and Tables S1-S3: the P-values of Tukey's tests in Table S1-S3 have been completely misunderstood in that P>0.05 are in bold and considered significant. The corresponding letters above the bars in Fig 2a & b are also wrong. Fig 2 caption line 101 should refer to Tables S1-S3.
The discussion is poorly structured. The initial emphasis on horticultural applications does not reflect the aim of the study. Correlation with flowering time is mentioned (line 176) but not presented in the results. The discussion does not reflect the conclusions in the abstract.
Minor comments
line 35: many, but not all, alpine species are able to reproduce clonally - please modify wording
line 47: change "probably imposed by" to "probably a consequence of"
line 52: change "has" to "have"
line 146: change to "Application of auxin to Tot.."
line 164: change to "delivery of auxin to intact.."
line 167: delete "Like in many species" - Wca and Dor dont occur in other species
lines 168-170. the last sentence of this paragraph would be better in line 162 after refs [30,31]
line 172: delete "as reported" as this has already been said
line 174: say "many alpine species are able to clonally .." as not all can clonally propagate and even the ones that can dont do so all the time
Fig 1 (a) change "internode" on legend to "main stem" to match text
Fig 3 & Fig 4 Y-axis labels: change "ratio" to "proportion"
line 230: change wording "was used to test for significant differences (P<0.05) .."
Reviewer 2 Report
The authors have presented a very thorough analysis of very difficult work to anayse genotype x developmental stage/age x treatment for adventitious root production and the effects of exogenous auxin on the adventitious root formation. The authors also quantified the free auxin levels and the expression of GH3 in each of the ecotypes.
This reviewer has a few questions that the authors should be able to easily clarify in the text.
Abstract
- It is not initially clear that Wca is an ecotype/accession for those reader unfamiliar with accession/ ecotype names in this species. Perhaps the authors might say Wca accession/ ecotype instead of Wca plants?
- The abstract states that Wca plants did not respond to NAA, but it seems that Dor plants did not respond to NAA from the results? Should that be included here and perhaps a qualification of the Wca NAA-resistance? It seems that Wca looks insensitive to NAA with respect to internodes, but responsive with respect to branching?
Results
It is very interesting that adventitious roots developed late (45 d) on the hypocotyl of pep1-1;, and the adventitious roots in hypocotyl vs internodes is also interesting. (just an observation, nothing to do)
- Do all Wca plants have adventitious roots at internode 3 (Fig S2), if I am reading the graph correctly? That is interesting. If so, can the authors speculate on this, or perhaps it’s just specific to this ecotype?
- If so, that is very interesting. Was that the internode in Wca (and other ecotypes) that was examined in Figure 2 for gene expression and free IAA levels?
- Is the Dor accession also insensitive to NAA?
Discussion
- Would that authors like to include a discussion of auxin and GH3 in the discussion? That would be nice for readers.
Materials and methods
- What was the light intensity/fluence?
A note to the journal (not the authors)
I liked that the figure legends were double spaced. It made them easy to read. It was a bit more challenging and took more time to review the paper since it was single spaced. Suggest journal have submission 1.5 or double spaced.
Round 2
Reviewer 1 Report
The authors have responded very well to my previous comments and addressed all of my concerns. The manuscript now reads very clearly and the amended discussion provides an appropriate context for the study.